# Transfer Learning for Evolving Domains

## Abstract

Transfer learning explores how to leverage knowledge from various tasks or domains (sources) to enhance predictive performance in related tasks or domains (targets). Typically, transfer learning research is segmented into several isolated sub-areas (such as domain generalisation, domain adaptation or multi-domain learning), each making distinct assumptions about target data availability, such as the quantity of data and labels available at training time. However, in many real-world applications, data availability is not fixed but evolves over time, as instances and labels are progressively collected from a new domain. Each of the classical settings then describes only a snapshot of a trajectory that a deployed system must traverse in full. We formalise this trajectory as a transfer learning problem in its own right, *Transfer Learning for Evolving Domains (TrED)*, specified by a data availability process fixed by the environment, a learning protocol that the method is free to choose, and an evaluation criterion that scores the whole trajectory of models rather than a single one. Within this formalism, the classical settings are recovered as regimes that a learner may pass through, rather than as separate problems that TrED concatenates. We then examine the transfer learning literature to identify mechanisms that are promising building blocks for a solution, and find that most methods are tailored to a single regime and that even the strongest existing candidates do not yet optimise the whole trajectory. We argue that TrED is a well-posed and unsolved problem, and an important direction for future research.

## 1 Introduction

Machine Learning (ML) models have become the state-of-the-art approach for various predictive tasks, such as classification and regression. However, the performance of these models relies heavily on the availability of data, which can be difficult and costly to obtain. For example, labelling an event for money laundering detection requires manual work from a domain expert, making the compilation of a large annotated dataset time-consuming and expensive (Barata et al., 2021).

One approach to address these challenges is Transfer Learning (TL), which aims to leverage knowledge from one or more tasks or domains (Source) to enhance performance in another task or domain (Target).[1] This covers a range of scenarios, each making different assumptions about the volume and type of data available from the source and target domains at training time. For example, some approaches may use a small amount of labelled data from the target domain, while others rely solely on labelled data from the source domain.

Traditionally, different TL scenarios are treated as distinct problems with specific solutions. However, in real-world applications, there are many cases where these scenarios are not fixed but evolve over time, as more data and labels are progressively collected from a new domain.

Consider global services such as a streaming platform (e.g., Netflix), an e-commerce site (e.g., Amazon), or a financial service (e.g., PayPal) expanding into a new country or region. Initially, there might be no user data from this new market, but they can use historical data from other geographies to design a first solution. As the service gains users, data from the new market is continuously collected, enabling the improvement and adaptation of the deployed system. One real-world example showing the importance of this adaptation was studied at Spotify, where it was found that "users behave and interact differently in different markets" and

---

[1]In this paper, we focus on the TL setting of having a single task and various domains.

as such it was more effective to switch to a localised model once enough data is available to train it (Roitero et al., 2020).

Another illustrative example of changes in the nature of data used by ML models are regulatory changes, which may not only change the data collection processes but also the behaviour of people and institutions governed by it. This can disrupt the distribution of features and/or labels for deployed ML models, leading to drops in predictive performance and the need to execute a model update. For example, due to the General Data Protection Regulation enforced by the European Union, companies may be required to restrict their data collection policies, which means that the deployed models would stop receiving several features that were used to train them (Sartor et al., 2020). However, as companies adapt to the new legal environment, they continue to collect and utilise data within the regulatory framework, leading to the development of new models that respect user privacy.

A similar progression can be seen during a new disease outbreak. In the initial absence of patient data, TL can leverage data from previous outbreaks or related diseases to develop early diagnostic models. As specific data for the new disease (such as symptoms and outcomes) accumulate over time, these models can be continuously updated and refined to improve their predictive accuracy. This strategy was recently employed during the COVID-19 pandemic. Early diagnostic models were developed using pre-trained computer vision models and data from similar infections. These models were then fine-tuned using the limited COVID-19 data available at the time (Altaf et al., 2021; Narin et al., 2021).

These examples have a common structure: multiple source domains provide a large pool of labelled data, and a new target domain is introduced with initially no data. As data collection begins, obtaining labels may be delayed, and over time the quantity of both instances and labels from the target domain increases. Figure 1 illustrates how this common data availability path overlaps with the typical settings of several classical TL sub-problems. Each such setting describes the target data availability at some point along this path, but none of them accounts for its evolution, and treating each snapshot in isolation misses the fact that a deployed system must traverse the whole path.

We study this evolution and formalise it as a new transfer learning problem, *Transfer Learning for Evolving Domains (TrED)*, in which data availability changes over time and a learner is judged on its performance throughout that evolution rather than at a single point of it. We give TrED a formal specification, show that the classical settings are recovered as regimes of it rather than as separate problems it concatenates, and examine the TL literature to identify which existing mechanisms are promising building blocks for a solution. As no single method currently addresses the whole trajectory, we highlight techniques that could be lifted from a single snapshot to spanning a broader range of it. From an application point of view, we anticipate that unified solutions, capable of effectively incorporating new target domain data, will speed up improvements in model performance and facilitate a more streamlined ML pipeline, reducing the need to frequently switch methodologies.

In Section 2 we introduce some notation and define the four most relevant classical TL sub-problems. In Section 3 we propose the new TrED problem, with a formal description of the data collection process, the learning protocol, and the evaluation metrics. In Section 4 we discuss promising paths towards a TrED solution. In Section 5 we present our conclusions and directions for future work.

## 2 Background

Transfer Learning (TL) is usually described as a set of methods to reuse knowledge from one task or domain to improve the performance on a different but related task or domain (Weiss et al., 2016). Before introducing our problem, we establish the necessary background: we first define the core concepts of "domains" and "tasks", and then describe the classical problem settings studied within TL.

### 2.1 Definitions and Notation

Following the definitions from Pan & Yang (2009), a **domain** $\mathcal{D} = \{\mathcal{X}, P(X)\}$ consists of a feature space $\mathcal{X}$ and a marginal probability distribution $P(X)$, where $X \in \mathcal{X}$ is a random variable representing the observed

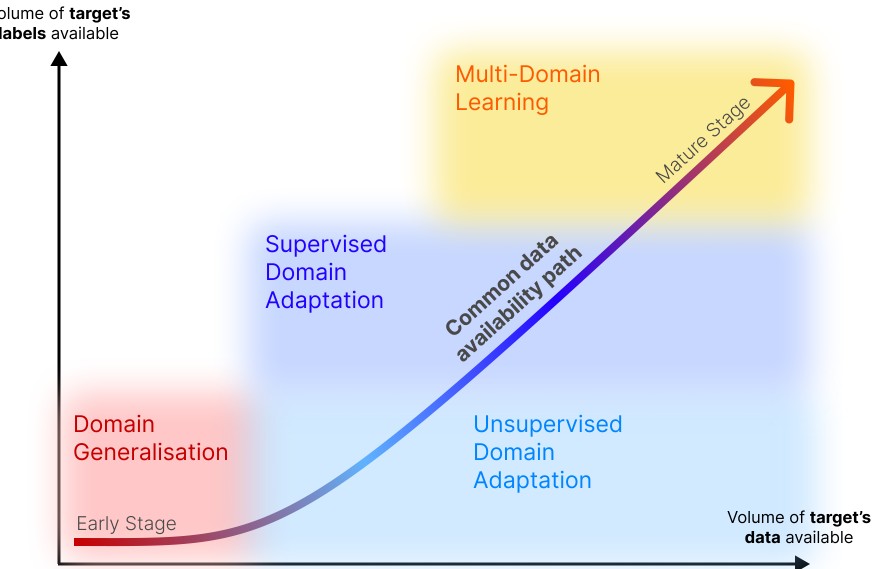

Figure 1: Transfer learning can be divided into different sub-problems that make different assumptions about target domain data and labels available. Once a new domain is included, the available data typically evolves over time according to the depicted arrow, starting from a Domain Generalisation setting (without target domain data), then moving towards Domain Adaptation (with target data but none or limited target labels), and later into Multi-Domain Learning (with considerable labelled target domain data).

instances. As such, two domains $\mathcal{D}_1 = \{\mathcal{X}_1, P(X_1)\}$ and $\mathcal{D}_2 = \{\mathcal{X}_2, P(X_2)\}$ differ if they either have different feature spaces ($\mathcal{X}_1 \neq \mathcal{X}_2$) or different marginal probability distributions ($P(X_1) \neq P(X_2)$).

TL literature (Pan & Yang, 2009; Zhang et al., 2019; Zhuang et al., 2020) often presents two equivalent interpretations for the concept of task: as a deterministic predictive function $f : \mathcal{X} \to \mathcal{Y}$, or as a conditional probability distribution $P(Y|X)$. We adopt the probabilistic interpretation to better account for label uncertainty, noise, and potential concept drift over time. Given a specific domain $\mathcal{D}$, a **task** $\mathcal{T} = \{\mathcal{Y}, P(Y|X)\}$ consists of a label space $\mathcal{Y}$ and a conditional probability distribution $P(Y|X)$, which represents the predictive relationship between features and labels.

One domain can encompass multiple tasks (e.g. using the same set of images for an image classification task or an object detection task), and different domains can share the same task (e.g. having a spam detection task on texts from different languages). The formal definition of task would require that tasks in different domains should have the same label space and conditional probability distribution (and thus the same feature space) in order to be considered equal. We relax this requirement by considering tasks in different domains to be equivalent ($\mathcal{T}_1 \sim \mathcal{T}_2$) if they involve solving the same underlying problem. For example, we can have a common task of image classification on two image domains, one in colour and one in greyscale, even though the domains have different feature spaces (3 colour channels versus 1).

**Transfer Learning** is the sub-field of ML that studies how to leverage datasets from various domains and/or from different tasks to learn a better performing model. More formally, given a source domain $\mathcal{D}_S$ with a learning task $\mathcal{T}_S = \{\mathcal{Y}_S, P_S(Y|X)\}$ and a target domain $\mathcal{D}_T$ with a learning task $\mathcal{T}_T = \{\mathcal{Y}_T, P_T(Y|X)\}$, where $\mathcal{D}_S \neq \mathcal{D}_T$ or $\mathcal{T}_S \neq \mathcal{T}_T$, TL aims to use the knowledge of $\mathcal{D}_S$ and $\mathcal{T}_S$ to learn a predictive function $f : \mathcal{X} \to \mathcal{Y}$ that is a better approximation of $P_T(Y|X)$ than what could be learned only from $\mathcal{D}_T$. It is possible to have various source domains and various target domains, and these two sets can be disjoint, have some overlap, or be the same.

In this work, we focus on the TL setting with different domains ($\mathcal{D}_S \neq \mathcal{D}_T$) but similar tasks ($\mathcal{T}_S \sim \mathcal{T}_T$).

Table 1: Data availability assumptions at training time from different TL settings: Domain Generalisation (DG), Unsupervised/Supervised Domains Adaptation (UDA / SDA), Multi-Domain Learning (MDL). In SDA, there may or may not be large volume of unlabelled target domain data. In MDL, there is no distinction between source and target domains, but there is a large volume of labelled data from all domains.

| TL Setting | Labelled Source Data | Unlabelled Target Data | Labelled Target Data |
|---|---|---|---|
| **DG** | $\gg 0$ | $= 0$ | $= 0$ |
| **UDA** | $\gg 0$ | $> 0$ | $= 0$ |
| **SDA** | $\gg 0$ | $\gg 0$ or NA | $> 0$ |
| **MDL** | $\gg 0$ | NA | $\gg 0$ |

## 2.2 Classical TL settings

Within the topic of TL, different settings have been described and addressed, making different assumptions about the datasets used to train the ML models. In this paper, we mainly focus on four of these settings, which we identify as those that align with real-world data availability conditions (Figure 1): Domain Generalisation, Unsupervised Domain Adaptation, Supervised Domain Adaptation, and Multi-Domain Learning. They all assume that a large pool of labelled source domain data is available at training time. Their main differences relate to whether target domain data is available and whether it is labelled or unlabelled. We summarize their assumptions in Table 1.

In **Domain Generalisation (DG)** (Zhou et al., 2022; Wang et al., 2022a), there is no target domain data available for training. The goal is to use the labelled data from source domains to learn a predictive function $f$ that is a good approximation of $P_T(Y|X)$ for any new target domain $\mathcal{D}_T$, unknown at training time.

In **Unsupervised Domain Adaptation (UDA)** (Wilson & Cook, 2020), there is only unlabelled target domain data available at training time. The goal is to use the labelled data from source domains and the unlabelled target data to learn a predictive function $f$ that approximates $P_T(Y|X)$ on $\mathcal{D}_T$.

In **Supervised Domain Adaptation (SDA)** (Wang & Deng, 2018), there is a small volume of labelled target domain data at training time, and there may be a much larger volume of unlabelled target domain data as well. The goal is to use the labelled source data and the small labelled target data (and unlabelled target data, if available) to learn a predictive function $f$ that is a good approximation of $P_T(Y|X)$ on $\mathcal{D}_T$.

In **Multi-Domain Learning (MDL)** (Yang & Hospedales, 2014), there are various labelled datasets from different domains available at training time, with no distinction between source and target domains. The goal is to use labelled data from all domains to learn one or multiple predictive functions such that the learned model(s) can perform well on each individual domain $\mathcal{D}_1, \ldots, \mathcal{D}_m$. Unlike DG, MDL has access to datasets from all domains (including the target) during training. Unlike Domain Adaptation, MDL aims to optimise performance across multiple domains simultaneously, rather than focusing on transferring knowledge to a single target domain.

## 3 Transfer Learning for Evolving Domains (TrED)

The four TL settings described in Section 2.2 share a common structure: each fixes a snapshot of data availability at a training time $t_{tr}$, and asks for the model that best exploits that snapshot. This is a reasonable idealisation of a static benchmark, but it is a poor description of a deployed system. In a production pipeline, a new domain does not arrive with a fixed budget of labelled and unlabelled data: it arrives empty, accumulates unlabelled instances at a rate set by usage, and accumulates labels at a rate set by an annotation process that may lag behind. The data availability conditions that the classical settings treat as *given* are, in reality, a *process*.

We call Transfer Learning for Evolving Domains (TrED) the transfer learning problem in which data availability evolves over time and the learner is judged on its performance throughout that evolution, rather than at a single point of it. TrED is not a new method, nor a composition of the four settings above: it is a

specification of a problem in which those settings appear as particular snapshots. An instance of TrED is characterized by three main aspects, which we develop in turn:

1. a **data-availability process** (Section 3.1), which describes how features and labels become available in each domain over time, and is a property of the environment rather than a choice of the method;

2. a **learning protocol** (Section 3.2), which specifies the model update strategy, constrained by the available data at each time step;

3. an **evaluation criterion** (Section 3.3), which scores a learner over the whole trajectory rather than at a single time.

Section 3.4 then shows that DG, UDA, SDA and MDL can be recovered by evaluating TrED at specific time intervals, and Section 3.5 distinguishes TrED from related learning paradigms.

## 3.1 Evolution of data availability

In the standard ML setting, datasets are static collections of observations used to estimate $P(X)$ and $P(Y|X)$. In real-world pipelines, however, data is collected from multiple domains over time, and labels may arrive after a delay that depends on the annotation process.

To formalise this evolving availability, we refine the definition of a dataset to include temporal information. Each domain $\mathcal{D}_d$ is associated with a dynamically growing dataset $D_d = \{(x_i, y_i, t_i^x, t_i^y) \mid i = 1, \ldots, n_d\}$, where $x_i \in \mathcal{X}_d$ is a feature vector, $y_i \in \mathcal{Y}_d$ is its label, $t_i^x$ is the timestamp at which $x_i$ is collected, and $t_i^y \geq t_i^x$ is the timestamp at which $y_i$ becomes available. The difference $t_i^y - t_i^x$ is the **label delay** of instance $i$, which may not be constant and whose distribution is a characteristic of the annotation process.

At any time $t$, $D_d$ can be decomposed into two disjoint subsets:

$$D_d^L(t) = \{(x_i, y_i) \mid t_i^y \leq t\},$$
$$D_d^U(t) = \{x_i \mid t_i^x \leq t < t_i^y\},$$

the instances whose labels have arrived by time $t$, and those that have been observed but whose labels are still pending. We write $D(t) = \{(D_d^L(t), D_d^U(t))\}_{d=1}^m$ for the full data availability state at time $t$.

We highlight two important properties of this formulation. First, the data availability process (i.e. the trajectory of $|D_d^L(t)|$ and $|D_d^U(t)|$ over time) is not determined by the learner, but by the deployment environment: how fast the domain is used, how fast annotators work, whether labels arrive from user feedback with a natural delay. The learner does not get to decide when it has target labels; it only gets to decide what to do with the ones it has.[2] Second, even though any fixed $t$ yields an availability state that some classical TL setting already describes, none of them consider the natural evolution of data availability, which could be leverage to better optimize the learner by having a more holistic view of this process.

## 3.2 Learning Protocol

While the availability process says what data exists, a learning protocol $P$ says what the learner does with it, and when. The learner selects a sequence of update times $t_1 < t_2 < \cdots < t_K$ within the deployment horizon $[t_{start}, t_{end}]$. For each $t_k$, it produces a model:

$$f_{t_k} = \arg \min_{f \in \mathcal{F}} \ell_k\big(f; D(t_k), f_{t_{k-1}}\big),$$

by minimising a per-step loss $\ell_k$ over the availability state $D(t_k)$, possibly depending on the previously deployed model $f_{t_{k-1}}$. The model $f_{t_k}$ becomes the *active* model (the one used to serve predictions) until it is replaced at $t_{k+1}$.

Several properties of this protocol are worth stating explicitly.

---

[2]This is what separates TrED from active learning, in which the acquisition of labels is precisely the decision variable.

- The update schedule $t_1, \ldots, t_K$ and the loss $\ell_k$ are decisions of the learner, not part of the problem statement. The update times don't need to be equally spaced or fixed in advance, and may be chosen adaptively as a function of the observed state.

- At each $t_k$, the accessible state $D(t_k)$ may contain instances whose labels have not yet arrived ($t_i^y > t_k$), which are available only as unlabelled data through $D_d^U(t_k)$. It is this gap between observation and labelling that leads to a regime in which unlabelled target data is available before its labels (the situation the UDA setting isolates).

- The entire past history is available at every update, not merely the interval $[t_{k-1}, t_k)$. Nothing forces the learner to discard earlier data, and it may retrain on the full accumulated state $D(t_k)$ for any $k$.

TrED does not specify $\ell_k$ or the update schedule, but rather identifies them as the open problems to be studied. As such, a method that retrains from scratch on all available data at fixed intervals is a valid TrED learner. So is a method that switches between off-the-shelf algorithms according to which classical setting the current state most resembles (Section 4.1), or a method that maintains a single continuously updated model, with a loss that anneals from a domain invariance objective towards a target specific objective, as target labels accumulate over time. TrED provides the terms in which these methods can be written down, with Section 3.3 describing how they can be compared.

### 3.3 Evaluation

The learning protocol produces not a model but a *trajectory* of models: the active model at time $t$ is $f_{\tau(t)}$, where $\tau(t) = \max\{t_k \mid t_k \leq t\}$ is the most recent update at or before $t$. Evaluating a TrED learner therefore means scoring a trajectory, and the choice of how to aggregate over it is a modelling decision that we now make explicit.

**Cumulative target risk.** The natural population-level criterion is the risk of the active model on the target domain, integrated over the deployment horizon:

$$R(P) \; = \; \frac{1}{t_{end} - t_{start}} \int_{t_{start}}^{t_{end}} \mathbb{E}_{(x,y) \sim P_T(X,Y)} \big[ \, \mathcal{L}\big(f_{\tau(t)}(x), \, y\big) \, \big] \; dt.$$

This is an evaluation criterion, not a training objective. It is defined over the true target distribution and the whole horizon, neither of which is accessible to the learner at any $t_k$. The relationship between $R(P)$ and the per-step losses $\ell_k$ that a learner can actually minimise is one that we identify as an open problem.

We also highlight that this criterion rewards solutions for improving sooner. A learner that reaches low target error quickly and holds it is preferred to one that reaches the same error only after the domain has matured, even though both are indistinguishable under the classical evaluation, which scores the final model alone. This allows one to measure the full cost of the trajectory and compare the paths that different TrED solutions may follow.

**Empirical estimation.** In practice $P_T(X, Y)$ is unknown and time is observed only through the collected instances, so $R(P)$ must be estimated from the data itself. The direct estimator averages the loss of the active model over the collected target instances, evaluated at the time each was observed:

$$\hat{R}(P) \; = \; \frac{1}{n_T} \sum_{i=1}^{n_T} \mathcal{L}\big(f_{\tau(t_i^x)}(x_i), \, y_i\big).$$

Note that each instance is scored by the model that would have served it.

This weights time by data density: periods of heavy traffic contribute more than quiet ones. This is often the desired behaviour, because errors are costly in proportion to how many predictions are made. But if the intent is to weight wall-clock time uniformly, the estimator should be re-weighted by the inverse local density of $t_i^x$, or the deployment horizon partitioned into equal windows, scored separately and averaged. This choice is specific to the problem or deployment requirements.

**Secondary criteria.** $R(P)$ measures error against time, but there are two other axes that are often relevant in deployment scenarios. The first is *performance at a data budget*, i.e. the target risk attained as a function of $|D_T^L(t)|$ rather than of $t$, which isolates label efficiency from collection speed. The second is *update cost*, i.e. the number of update times $K$ times the compute expended per update, since a learner that retrains from scratch on every incoming label may attain excellent $R(P)$ but be unfeasible for the real-world deployment settings. We do not fix a single figure of merit combining these, and we note that the absence of an agreed one is itself part of the open problem.

**Evaluation under label delay.** A subtlety that has no analogue in the classical TL settings: because $y_i$ may arrive at $t_i^y > t_i^x$, the term $\mathcal{L}(f_{\tau(t_i^x)}(x_i), y_i)$ is not computable at time $t_i^x$. The retrospective estimate $\hat{R}(P)$ is well-defined once the horizon is complete and all labels have arrived, and is therefore adequate for offline comparison of methods. But it is not necessarily available to a learner that wishes to monitor its own current performance, since the labels that would reveal it are still in flight. Any component of a method that requires an estimate of current target performance (e.g. an early-stopping criterion, or a model selection step) must therefore operate on a delayed or surrogate signal. We flag this now as an added difficulty of TrED, and we discuss how it affects the stage-switching family of solutions in Section 4.1.

### 3.4 Classical TL settings as regimes of TrED

We now recover the four classical settings. Consider an instance of TrED in which a new target domain $\mathcal{D}_T$ appears at $t_{start}$ with no data ($D_T^L(t_{start}) = D_T^U(t_{start}) = \emptyset$), while the source domains are already mature ($|D_{S_d}^L(t_{start})| \gg 0$). As the availability process runs, the state $D(t)$ passes through the following regimes.

- For $t \in [t_{start}, t_a)$, target instances are too few to estimate $P_T(X)$: $|D_T^U(t)| \approx 0$ and $|D_T^L(t)| = 0$. A learner updating during this interval faces the **DG** problem, with $D_{S_d}^L(t)$ playing the role of $D_{S_d}^L(t_{tr})$.

- For $t \in [t_a, t_b)$, unlabelled target instances have accumulated but most of their labels are lagging behind: $|D_T^U(t)| > 0$, $|D_T^L(t)| \approx 0$. A learner updating during this interval faces the **UDA** problem, with $D_T^U(t)$ playing the role of $D_T^U(t_{tr})$.

- For $t \in [t_b, t_c)$, the earliest labels have arrived while instances continue to accumulate: $|D_T^U(t)| \gg |D_T^L(t)| > 0$. A learner updating during this interval faces the **SDA** problem.

- For $t \geq t_c$, labelled target data is abundant, $|D_T^L(t)| \gg 0$, and the target is no longer significantly different from a source. A learner updating during this interval faces the **MDL** problem over $\{\mathcal{D}_{S_1}, \ldots, \mathcal{D}_{S_m}, \mathcal{D}_T\}$.

The transition times $t_a, t_b, t_c$ are not constants. They are induced by the availability process (i.e. the velocity of data collection and the label delay distribution) and they are not observable to the learner, since observing $t_b$ would require knowing that enough labels have arrived to make supervised adaptation worthwhile, which is a statement about target risk and therefore subject to the delay problem of Section 3.3. Also, these boundaries are not sharp. Nothing in the availability process changes discontinuously at $t_a$, and the interval around it is a continuum of states that are neither clearly DG nor clearly UDA. We are describing this simplified transition between problems to paint a mental picture of their relation to TrED, but real deployment scenarios are more nuanced for the reasons that we discussed.

As such, these classical settings can be seen as *regimes* of TrED, rather than components of it. Each is a description of the availability state at some times, and each comes with a literature of methods that exploit that state well. But TrED is not the concatenation of those four problems, and a TrED method does not need to recognise them necessarily, as long as it produces a trajectory of models with low cumulative risk. Whether the best way to do so is to detect these regimes and act accordingly is a question we take up in Section 4.1, where we find the answer is not necessarily yes.

### 3.5 Relation to other learning paradigms

TrED is related with several established paradigms, so it is worth to explicitly describe their similarities and differences. In what follows, recall the three components described in Section 3: an exogenous data availability process, a learning protocol governing data access and update schedules, and an evaluation criterion to compare different model trajectories.

**Online learning.** Online learning (Hoi et al., 2021) also produces a trajectory of models scored cumulatively, and its regret criterion is a close relative of $R(P)$, but with two main differences. First, online learning is single-stream: there is one distribution, and the challenge is sequential estimation, not transfer. Second, its protocol is typically restrictive in a way that TrED's is not: the learner sees each example once, updates, and moves on. In TrED the learner retains everything, may retrain on the full history at any $t_k$, and is limited by the availability of labels rather than by memory or by a one-pass constraint. Online techniques are nonetheless useful *within* TrED, as one way of implementing the update rule.

**Continual learning.** Continual learning (De Lange et al., 2021; Van de Ven et al., 2022) likewise concerns a model that evolves over a sequence of distributions. Its defining difficulty, however, is catastrophic forgetting, which arises from the assumption that past data is no longer accessible. TrED makes the opposite assumption: source data persists, and the historical state is revisitable, so forgetting is not forced by the protocol. Consequently the two problems reward different things: continual learning asks how to retain performance on past domains without their data; TrED asks how to reach performance on a *new* domain quickly, with its data arriving under delay. Methods from the continual learning literature remain relevant (especially when retraining from scratch is infeasible), but they solve a constraint that TrED does not impose.

**Online transfer learning.** OTL (Zhao et al., 2014) is the closest of the streaming paradigms: target data arrives sequentially and source knowledge is available. It differs from TrED in two aspects. First, the source domain knowledge is restricted to pre-trained models rather than data, while in TrED source data is always available and may be re-used. Second, the protocol is restricted to online, single-pass consumption of the target stream, while a TrED learner may retain and revisit the full history of target domain at any $t_k$. Notably, OTL does not specify which target regime it operates in (i.e. target data may be labelled or unlabelled). As such, it can be seen as a TrED protocol constrained to models-only source access and online updates, and we regard OTL methods as directly relevant building blocks.

**Test-time adaptation.** TTA (Liang et al., 2025) restricts source knowledge in the same way as OTL (the source domains are available only as a pre-trained model, not as data), but constrains the target axis rather than the learning protocol. Like UDA, it assumes the target instances used for adaptation are unlabelled; unlike UDA, it forbids access to the source data those instances are being adapted from. The protocol itself is left open: adaptation may happen offline, once, on a batch of target data (the setting also called Source-Free Domain Adaptation (Li et al., 2024b)), or online, on each test batch as it arrives (Wang et al., 2020; 2022b). Thus, both OTL and TTA restrict access to source data (only models), but the former fixes the online learning protocol and accepts different target regimes, while the latter specifies unlabelled-only target data and admits either update protocol. In this sense, TTA can be read as one regime of TrED (the unlabelled-only target interval) with an additional source-access restriction.

**Gradual domain adaptation.** GDA (Kumar et al., 2020) shares the TrED intuition that domains evolve over time, but in GDA the domain itself is shifting: the data distribution drifts through a sequence of intermediate domains by design, and the main insight is that self-training along that sequence of domains can lead to better transfer performance than when trying to cross a large gap between the domains directly. In TrED, the target domain is fixed, while its data availability is evolving over time. Therefore, the two settings are complementary: GDA answers what to do when $P_T$ drifts, TrED asks what to do while $D_T^L$ and $D_T^U$ are being collected, and a fully realistic deployment exhibits both. We hold $P_T$ fixed throughout this paper in order to isolate the availability axis, and we regard their combination as a natural extension rather than as within our present scope.

# 4 Paths to a TrED solution

Having defined TrED as a problem, we now ask how it might be solved. We aim to map the space in which a solution would live, and highlight promising paths to explore in future work.

In Section 4.1, we examine a simple potential approach: to detect which classical regime the current state resembles and deploy the corresponding off-the-shelf method. We explain why, despite being a reasonable baseline, it is neither trivial to run nor obviously optimal. We then turn to the existing TL literature in Section 4.2, to ask which popular mechanisms are promising ingredients for a TrED learner, under which data-availability conditions each is most valuable, and what would have to change to lift them from a single snapshot to the whole trajectory. Section 4.3 highlights some examples of methods that already encompass a broader range of the TrED continuum (e.g. methods that already propose how to integrate some target labels into UDA). Lastly, Section 4.4 describes foundation models as the strongest existing candidate for the basis of a complete TrED solution, mapping its capabilities onto data availability regimes and highlighting the limitations that still need to be addressed.

## 4.1 The stage-switching baseline

The most immediate way to attack TrED is to reduce it to the problems we already know how to solve. Section 3.4 shows that the availability state passes through DG, UDA, SDA and MDL regimes as target data accumulates. As such, the obvious response is to keep one method for each regime and switch between them as the state crosses the transition times $t_a$, $t_b$, $t_c$. We call this the **stage-switching baseline**. It is a legitimate TrED learner in the sense of Section 3.2: it selects update times and a per-step loss, here by delegating each interval to a specialised algorithm. And it is a strong baseline, because each constituent method is the product of a mature literature that studies that particular regime.

Nevertheless, this type of solution has two main limitations that may lead to it having worse performance than a TrED-specific solution.

The first is that the switch is hard to time. As mentioned in Section 3.4, the transition times described as $t_a$, $t_b$, $t_c$ are simplifications of what is in reality a continuous process. Even if we define them in terms of performance maximization (i.e. $t_b$ signals that enough target labels have arrived for supervised adaptation to beat unsupervised adaptation), due to the label delay referred in Section 3.3, the true target risk is not necessarily observable at decision time. The switcher must therefore commit to a switch on a proxy (an instance count, a fixed calendar schedule) that may not track the quantity that actually determines which method is better. A mistimed switch can be worse than not switching: adopting an SDA method on a handful of noisy early labels can underperform the UDA method it replaces. An oracle that knew the transition times could switch optimally between the four regime-specific solutions, but a deployed learner cannot, and closing that gap is itself an open sub-problem.

The second, and deeper, problem is that switching presupposes a specific decomposition. A switching pipeline can only express solutions of the form "run method $A$, then method $B$, then ...": a piecewise-constant policy over a partition of the timeline into named regimes. But nothing about TrED guarantees the optimum has this form. The cumulative-risk criterion of Section 3.3 rewards the whole trajectory, and the best trajectory may be produced by a single model whose objective changes continuously as target data and labels accumulate (e.g. a loss that anneals from a domain-invariance objective towards a target-specific objective). One might hope to recover this by relaxing the hard switch into a smooth weighting of the four regime solutions, with weights that depend on the data available. This widens the set of expressible trajectories but does not close the gap: the four solutions form a fixed basis, and a model that lies outside their span cannot be reached by any weighting of them, regardless of how the weights vary with the state.

We therefore treat the stage-switcher as what it is: a greedy, oracle-dependent baseline that any serious method must beat. We argue that producing a trajectory of lower cumulative risk than the best stage-switching baseline is a well-posed problem, and to our knowledge an unsolved one.

Table 2: Reviewed TL methods, organised by the type of technique they employ.

| Category | Technique | Methods |
|---|---|---|
| Data Transform. | Instance Weighting | Dai et al. (2007); Sun et al. (2011); Wang et al. (2017); Yao & Doretto (2010) |
| | Feature Mapping | Muandet et al. (2013); Panareda Busto & Gall (2017); Saenko et al. (2010); Sun et al. (2016) |
| | Pseudo-labelling | Berthelot et al. (2021); Chen et al. (2011); Li et al. (2021); Nguyen et al. (2025); Panareda Busto & Gall (2017); Qiu et al. (2021); Saito et al. (2017); Sener et al. (2016); Singh (2021); Sun et al. (2025); Wang et al. (2017; 2022b;c); Yuan et al. (2023) |
| | Feature expansion | Daumé III (2009); Duan et al. (2012); Kumar et al. (2010) |
| Model Adaptation | Modular DL (a) | Dou et al. (2019); Ganin et al. (2016); Ghifary et al. (2016); Li et al. (2018b); Long et al. (2018); Saito et al. (2018; 2019); Sun et al. (2019; 2025); Wang et al. (2022c) |
| | Modular DL (b) | Bao et al. (2019); Li et al. (2021); Long et al. (2015); Nam & Han (2016); Peng et al. (2019); Sener et al. (2016); Singh (2021) |
| | Modular DL (c) | Tzeng et al. (2017); Zhong et al. (2022) |
| | Modular DL (d) | Bousmalis et al. (2016); Chen & Cardie (2018); He et al. (2023b); Wu & Guo (2020); Xi et al. (2024) |
| | Ensemble | Dredze et al. (2010); Li et al. (2021); Peng et al. (2019); Saito & Saenko (2021); Vu et al. (2022); Zhong et al. (2022); Yang et al. (2024); Zhou et al. (2025) |
| Learning Objective | Adversarial Loss | Bousmalis et al. (2016); Chen & Cardie (2018); Ganin et al. (2016); He et al. (2023b); Hoffman et al. (2018); Li et al. (2018b; 2021); Long et al. (2018); Nguyen et al. (2025); Saito et al. (2018; 2019); Sun et al. (2025); Tzeng et al. (2017); Wu & Guo (2020); Yuan et al. (2023); Zhou et al. (2020) |
| | Contrastive Loss | He et al. (2023b); Huang et al. (2022); Singh (2021); Sun et al. (2025); Wang et al. (2022c); Yuan et al. (2023) |
| | Meta Learning | Dou et al. (2019); Gao et al. (2022); Li et al. (2018a; 2019); Li & Hospedales (2020); Liu et al. (2020); Qiu et al. (2021); Zhong et al. (2022) |
| | Reconstruction Loss | Bousmalis et al. (2016); Ghifary et al. (2016); He et al. (2023a); Hoffman et al. (2018); Li et al. (2018b); Lin & Xue (2025) |
| | Statistical Distance | Bousmalis et al. (2016); Dou et al. (2019); Garg et al. (2022); He et al. (2023a); Li et al. (2018b); Long et al. (2015; 2016; 2017); Nguyen et al. (2025); Peng et al. (2019); Tzeng et al. (2014) |

## 4.2 Building blocks from the TL literature

The four classical TL settings described in Section 2.2 have accumulated a large body of work tailoring learners to their particular data availability assumptions. Given that a simple composition of pre-existing methods has the limitations highlighted in Section 4.1, we instead turn to the specific techniques that they use, asking how each might serve as a building block for a TrED-specific solution. We organise the discussion by technique rather than by setting, and for each we ask what data availability it requires, where on the trajectory it is most valuable, and what would have to change to broaden its range. We group techniques into three families according to which component of the ML pipeline they act on: the training data, the model, or the learning objective. These axes are largely independent, and several methods combine techniques from more than one group. Table 2 maps the reviewed methods onto these families.

### 4.2.1 Data transformation

Data transformation techniques modify the training data before it reaches the model.

**Instance weighting and feature mapping** share the same goal: aligning the source and target feature distributions, without affecting the objective or the model architecture. The difference is where they intervene. Instance weighting leaves the data untouched, and instead weights source examples such that the reweighted source distribution resembles the target's (Sun et al., 2011; Wang et al., 2017). Feature mapping leaves the sampling untouched and reshapes the representation, transforming the feature space so that the mapped source and target distributions are better aligned (Muandet et al., 2013; Panareda Busto & Gall, 2017; Saenko et al., 2010; Sun et al., 2016). They also have different update costs (as defined in Section 3.3): reweighting is cheap to update, since new target data only re-estimates the weights, whereas a learned mapping must typically be re-fitted. Most of these methods align the marginal distribution of features, and as such can be used in TrED as soon as unlabelled target data becomes available. But some methods require target labels, to correct for differences that the marginal cannot capture (e.g. differences in the conditional distribution $P(Y|X)$). One notable example is TrAdaBoost (Dai et al., 2007; Yao & Doretto, 2010) that down-weights source instances that hurt target performance, which implicitly accounts for conditional differences. The idea of shifting from aligning marginal distributions to aligning conditional distributions as target labels accrue can potentially be implemented as an instance weighting policy for TrED.

**Pseudo-labelling** uses a model to create labels for unlabelled data, and then trains on the result. It is a very common technique wherever labels are scarce: in UDA, it supplies the missing target labels outright (Chen et al., 2011; Li et al., 2021; Nguyen et al., 2025; Qiu et al., 2021; Saito et al., 2017; Sener et al., 2016; Sun et al., 2025; Wang et al., 2017; 2022b;c; Yuan et al., 2023); and in semi-supervised SDA, it fills the gap between the few real target labels and the many unlabelled instances (Berthelot et al., 2021; Panareda Busto & Gall, 2017; Singh, 2021). For TrED, this technique has two desirable properties. First, it can directly address label delay, because a pseudo-label serves as an estimate of the label that has not yet arrived for a recently observed instance. Second, a pseudo-labelling learner can gracefully transition to real labels as they arrive: because real and generated labels can enter the same loss, the learner shifts the mixture from pseudo to real as $D_T^L(t)$ fills, with no change to the model architecture or learning objective. Therefore, it spans the UDA-SDA transition as a continuous shift in the label mixture rather than a discrete switch. The main limitation is fidelity: a pseudo-labelling step is only as good as the labels it creates, and early in a domain's deployment horizon, when the model is weakest, is exactly when the labels are least reliable. As such, a TrED learner needs a confidence (Berthelot et al., 2021) or agreement (Saito et al., 2017) criterion, to avoid training on its own errors.

**Feature-space expansion** replicates the feature space into source-specific, target-specific, and shared copies (Daumé III, 2009; Duan et al., 2012; Kumar et al., 2010). This allows the model to learn separately from the parts that transfer across domains and the parts that are private to one. Because the expansion is a preprocessing step on the representation, it composes with any downstream model, and can be applied from the DG regime onward, before any target data exists. Its limitation for TrED is that the representation grows with the number of domains, so it does not scale to a setting where domains accumulate indefinitely. But the intuition to separate shared transferable knowledge from private domain-specific knowledge is still very relevant for TrED, and is the same intuition that the modular architectures of the next section implement in a more scalable way, by locating the shared-versus-private distinction in the model rather than in the data.

### 4.2.2 Model adaptation

Model adaptation techniques design specialized architectures that facilitate the transfer of knowledge across domains.

**Modular Deep Learning** splits the network into feature extractors (that map inputs to embeddings) and classifiers (that map embeddings to predictions), with variations on which components are shared across domains and which are private to one. Shared components are used to make predictions in any domain, while private components are replicated for every domain and each copy is trained independently with data from its corresponding domain. Figure 2 depicts some recurring designs. The shared-shared architecture (Figure 2(a)) makes the fewest demands on target data, and therefore is the most portable across the data availability spectrum of TrED: it can be trained from source data alone and used in the DG regime, then continued

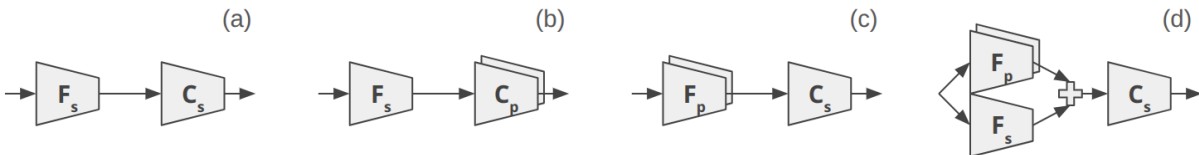

Figure 2: Diagrams of four different types of modular deep learning architectures used in transfer learning methods. The $F$ components are feature extractors, $C$ are task classifiers, the subscripts $s$ and $p$ indicates whether the component is shared or private across different domains. On panel (d), the outputs of the feature extractors are concatenated before being passed to a shared classifier.

into UDA and SDA as target data arrives. The shared-private architecture (Figure 2(b)) allows the model to specialize its predictions to each individual domain (which may be desirable in domains with significant concept drift), but requires labelled data from each domain to fit its private component, so it is only applicable for mature domains in TrED ($|D_d^L(t)| \gg 0$). The private-shared architecture (Figure 2(c)) is especially useful for applications with heterogeneous domains (with different feature spaces), and, provided the private extractors are trained to align their outputs, it can mitigate the need for target labels in TrED settings with large label delay, since the shared classifier can be reused without retraining (though predictive performance hinges entirely on the quality of that alignment). The hybrid-shared architecture (Figure 2(d)) combines both a shared and various private feature extractors for additional expressive power and may provide a good balance between decent early-deployment performance (using the shared feature extractor) and competitive late-stage performance (using a private feature extractor), especially in settings with significant covariate shift across domains. These different designs highlight a clear pattern: shared components are more lenient regarding data requirements, but private components may have better performance in specific conditions (e.g. data drift across domains). As such, a promising solution for TrED is to start from the shared-shared design (the most portable one), and add private capacities incrementally only if the domains require them.

**Ensembles**   address the multi-source case by training a separate model per source and combining their predictions (Dredze et al., 2010; Li et al., 2021; Peng et al., 2019; Saito & Saenko, 2021; Vu et al., 2022; Zhong et al., 2022; Yang et al., 2024; Zhou et al., 2025). Their appeal for TrED is incrementality: a new domain can enter as a new member without retraining the others, which keeps the per-step update cheap in the sense of Section 3.3. The combination rule is the main knob to tune the ensemble at each step in the data availability continuum. Techniques such as meta-learning can be used to optimize a solution without access to target data, or to adapt it in a single batch (Zhong et al., 2022). Once unlabelled target data becomes available, techniques such as confidence-weighting can guide the combination of ensemble members (Dredze et al., 2010). With access to few target domain labels, even when training a full model from scratch is unfeasible, simple models such as linear combinations can be trained to directly optimize target performance (Li et al., 2021). For mature domains in TrED, the source members can act as a regularizer on the target model, constraining it where target data is still thin. The main drawback is that ensembles grow with the domain count in both memory and inference, which the update-cost criterion of Section 3.3 penalises directly. As such, a TrED ensemble would need a policy for retiring or merging members as domains accumulate, perhaps by consolidating mature domains into a shared model while keeping separate members only for domains that still benefit from them.

### 4.2.3   Learning objective

Learning objectives shape the loss rather than the data or the architecture. They have a high potential for being adapted to TrED because a loss term is naturally modular: it can be added, removed or re-weighted without redesigning the model, which is what the per-step loss $\ell_k$ of Section 3.2 calls for.

**Distribution alignment losses**   pull source and target representations together. One option is to directly minimize a statistical distance between embeddings from different domains, such as MMD (Li et al., 2018b; Long et al., 2015; 2016; 2017; Tzeng et al., 2014) or the KL divergence (Dou et al., 2019; Garg et al.,

2022; Nguyen et al., 2025). Another option is to train a domain discriminator adversarially with the feature extractor (Bousmalis et al., 2016; Ganin et al., 2016; Hoffman et al., 2018; Long et al., 2018; Saito et al., 2018; Tzeng et al., 2017). These techniques usually require target domain data to estimate the target distribution, and as such activate from the UDA regime on. But there are also some examples of methods (Li et al., 2018b; Zhou et al., 2020) that use these losses as a regularization term, in order to improve performance on any unseen domain, which means they can be applied from the onset of a new domain in TrED. The main drawback from these losses is that matching the feature marginals is blind to class identity, which means that it may inadvertently merge classes into each other (for example, if the class proportions differ across domains). Conditioning the alignment on (pseudo-)labels (Long et al., 2018; Nguyen et al., 2025; Sun et al., 2025) avoids this by aligning each class to itself rather than the distributions in aggregate, at the cost of depending on label estimates that may be unreliable early on.

**Meta-learning** objectives optimise not for the training domains but for fast adaptation to a held-out one (Dou et al., 2019; Li et al., 2018a; Li & Hospedales, 2020; Zhong et al., 2022). This makes them directly applicable for TrED: a criterion that explicitly rewards rapid adaptation to a new domain is well aligned with cumulative risk over a domain's early life. The main limitation is the well-known instability and expense of the nested optimisation, which the update-cost criterion again makes relevant.

**Self-supervised losses** are typically used as auxiliary terms to provide additional training signal, mostly from unlabelled data. These can include contrastive loss (Huang et al., 2022; Yuan et al., 2023), entropy minimisation (Long et al., 2016; Saito et al., 2019; Saito & Saenko, 2021), or reconstruction loss (Bousmalis et al., 2016; Ghifary et al., 2016; Hoffman et al., 2018; Li et al., 2018b; Lin & Xue, 2025). It is also common for contrastive losses specifically to be computed from pairs of instances from different domains that share the same (pseudo-)label (He et al., 2023b; Singh, 2021; Wang et al., 2022c), which is another way to implement class conditional alignment. Their value for TrED is that they keep a model learning early after the domain's deployment, while there is little target domain data available. But even for later regimes, these losses can easily compose with others to provide extra signal and structure for training. As such, they are best understood as a complement rather than a complete solution.

The recurring theme across these three families is that the mechanisms which transfer best to TrED are the ones that degrade gracefully as data availability changes: pseudo-labelling that absorbs real labels as they arrive; shared backbone architectures that do not need per-domain data; and annealable loss terms that shift emphasis without a discrete switch.

## 4.3 Methods that adapt to different regimes

Some authors include specific remarks on how to adapt their methods to different data availability requirements. Two pseudo-labelling methods (Berthelot et al., 2021; Panareda Busto & Gall, 2017) were proposed to tackle the UDA problem, but they mention that, if target labels are available (i.e. moving to SDA problem), they can replace their pseudo-labels counterparts and the remaining mechanisms of the solution remain the same. Likewise, three other UDA methods (Li & Hospedales, 2020; Long et al., 2015; Sun et al., 2011) that use labelled source domain data to train directly refer that, if target labels are available, an additional supervised loss term should be added to leverage that information.

Even though each adaptation is individually simple, their pattern is informative. First, the two techniques used (pseudo-labelling and additional loss) are also the ones that we identified as having the most potential for TrED, which corroborates our assessment. Second, they serve as examples that other authors already find it natural to make one method serve two adjacent regimes rather than treating them as separate problems requiring separate solutions. However, these adaptations all concern a single transition, from unlabelled to labelled target data, which is the boundary where the benefit is most immediate. The fact that the other transitions along the continuum are rarely addressed is precisely the reason why TrED remains an open problem.

## 4.4 Foundation Models

The mechanisms discussed so far are building blocks, each promising for part of the TrED trajectory but none complete on its own. Foundation models (FMs) are the strongest existing candidate for a single system that covers the whole trajectory: large-scale pre-trained models that serve as a general-purpose basis for many downstream tasks (Bommasani et al., 2021), and whose broad prior knowledge lets them operate, in some form, under every data-availability regime. We first summarise the FM landscape, then map their capabilities onto the regimes of TrED, and finally argue why even FMs do not yet constitute a TrED solution.

### 4.4.1 Current literature on FMs

Large language models (LLMs) are a type of FM for natural language processing (NLP), usually based on the transformer architecture (Vaswani et al., 2017) and trained with a self-supervised learning task. Examples of this type of model include the GPT family (Radford et al., 2018; 2019; Brown et al., 2020), BERT (Devlin et al., 2018), LaMDA (Thoppilan et al., 2022), OPT (Zhang et al., 2022) and PaLM (Chowdhery et al., 2023). These models are capable of capturing nuanced language patterns and contextual information, making them highly effective for tasks such as text classification, summarisation, translation, and conversational AI.

In computer vision (CV), models pre-trained on large image datasets like ImageNet (Deng et al., 2009) have become the standard for feature extraction in tasks like image classification or object detection. Different architectures have been proposed, such as Inception Network (Szegedy et al., 2015), ResNet (He et al., 2016), DenseNet (Huang et al., 2017) or EfficientNet (Tan & Le, 2019). More recently, generative models more similar to LLMs have also been rising in popularity, such as the DALL-E family (Ramesh et al., 2021; 2022; Betker et al., 2023) and Stable Diffusion (Rombach et al., 2022). These models can generate detailed and complex images from text descriptions (demonstrating their comprehensive representations of real-world concepts) and they exhibit the ability to generalise to unseen images and apply these abstractions in new and varied contexts.

Given their success in NLP and CV, other FMs have been developed for different use cases, including (but not limited to) tabular data (Guo et al., 2025; Hollmann et al., 2022; Kim et al., 2024; Thomas et al., 2024; Wang & Sun, 2022; Zhang et al., 2023a), time series (Das et al., 2023; Garza & Mergenthaler-Canseco, 2023), audio (Borsos et al., 2023; Vyas et al., 2023; Yang et al., 2023), medicine (Khare et al., 2021; Li et al., 2024a; Tu et al., 2024; Zhang et al., 2023b) and finance (Skalski et al., 2023; Wu et al., 2023). These specialised foundation models are tailored for specific domains, providing strong baselines that can be incrementally updated as new data becomes available. Some methods (Gardner et al., 2024; Hegselmann et al., 2023) have also explored adapting NLP FMs to process tabular data, demonstrating how to leverage the rich context of LLMs to interpret the meaning of column names and cell values.

Some FMs have been proposed that are multi-modal, meaning that they can receive and process different types of data. Early examples include CLIP (Radford et al., 2021) and ALIGN (Jia et al., 2021), which use separate encoders for image and text data, trained jointly using contrastive objectives to align representations in a shared embedding space. Later, other generative transformer-based architectures were proposed, such as Flamingo (Alayrac et al., 2022), PaLM-E (Driess et al., 2023) and GPT-4 (Achiam et al., 2023). More recently, native multi-modal models have emerged, trained end-to-end on diverse data types including text, images, audio, video, and code. Examples include GPT-4o (OpenAI et al., 2024), Gemini (Team et al., 2023), Claude 3 family (Anthropic, 2024), and Llama 3 family (Grattafiori et al., 2024). Combining modalities enables the model to learn richer representations, leading to a more robust understanding of underlying concepts, which ultimately enhances generalisation across tasks (Sarfraz et al., 2024).

### 4.4.2 Applying FMs on TrED

The reason FMs are compelling for TrED is that a single pre-trained model exposes a different adaptation mechanism at each level of data availability, so in principle one system can traverse the whole trajectory without being swapped out. We map these mechanisms onto the regimes of Section 3.4.

In the DG regime, before any target data exists, the zero-shot inference capabilities of FMs can act as a robust baseline for the target domain, owing to their massive and diverse pre-training. Since a new task

with no data must be "explained" to the model, usually in natural language, prompt engineering becomes the main lever for improving predictive performance and robustness (Brown et al., 2020).

In the UDA regime, once unlabelled target data is available, the FM can generate pseudo-labels for a smaller specialised model, distilling its knowledge into a cheaper model with lower inference latency (two of the FM's main deployment limitations). This is the same pseudo-labelling mechanism discussed in Section 4.2, with the FM playing the role of the label generator.

In the SDA regime, the few available target labels can be supplied directly in the prompt for few-shot in-context learning (ICL), letting the model adapt without any retraining (Dong et al., 2022). The limitation is the context size: as labelled data grows, it may no longer fit inside the prompt.

This limitation becomes worse in the MDL regime, when the accumulated target labels exceed the context window. The model must then move from in-context adaptation to parameter adaptation, either by fine-tuning (full or parameter-efficient, e.g. adapters over a frozen backbone) or by offloading history to Retrieval-Augmented Generation (Lewis et al., 2020), which pulls relevant past target data dynamically rather than holding it in context.

**Why FMs are not yet a TrED solution.** Even though FMs can operate in every regime, this is not the same as optimizing the entire trajectory, which is the goal of TrED. Four gaps remain. First, the mechanisms described above are still a set of regime-specific tactics: nothing specifies when to move from prompting to distillation or from ICL to fine-tuning, so an FM-based learner inherits the same transition-timing problem as the stage-switcher of Section 4.1, only with the transitions internalised. Second, the mechanisms above use unlabelled target data only to transfer knowledge the FM already has (e.g. distilling zero-shot predictions into a smaller model), but none of them includes a self-training or adaptation step that mines the unlabelled instances for signal the pre-trained model does not already contain. Third, each mechanism optimises the current step in isolation, whereas the cumulative criterion $R(P)$ rewards the whole trajectory, so an FM tuned greedily at each regime need not produce a low-cumulative-risk path (again, mirroring the stage-switcher argument from Section 4.1). Fourth, and specific to FMs, is evaluation leakage: because the target data may have appeared in the pre-training corpus, strong zero-shot or few-shot numbers can reflect memorisation rather than transfer, and any honest TrED evaluation of an FM must establish that the target domain was genuinely unseen. FMs are therefore the most capable starting point we have, but turning that capability into a TrED solution (one that decides its own transitions, leverages unlabelled data, and optimises the cumulative objective) remains an open problem.

## 5 Conclusion

In this paper, we study the subject of transfer learning, exploring methods that share knowledge across different domains. We describe how this subject is traditionally segmented into distinct static sub-problems, such as Domain Generalisation, Unsupervised and Supervised Domain Adaptation, and Multi-Domain Learning, each making different assumptions regarding data availability. However, in real-world applications, data and labels from new domains are often collected gradually over time, presenting a more dynamic challenge that no single one of these settings captures.

We propose Transfer Learning for Evolving Domains (TrED), the transfer learning problem in which data availability evolves over time and a learner is judged on its performance throughout that evolution rather than at a single snapshot of it. We give it a formal specification: a data-availability process fixed by the environment, a learning protocol that the method is free to choose, and a cumulative evaluation criterion that scores the whole trajectory of models. Within this formalism, the classical settings are recovered as *regimes* that a learner may pass through, rather than as separate problems that TrED concatenates.

Building on this formulation, we study the transfer learning literature to identify promising mechanisms for a TrED solution. We find that most methods are constrained to a single regime, and that even the strongest existing candidates, including foundation models, do not yet optimise the whole trajectory. Thus, our claim is that TrED is a well-posed and unsolved problem: one can state precisely what a solution is and how two candidate solutions are compared, and, to our knowledge, no existing solution satisfies all its demands.

By framing transfer learning as an evolving process rather than a collection of isolated snapshots, this perspective paves the way for more adaptive, resilient models capable of improving over time. We believe this approach will contribute to creating robust systems suited to real-world scenarios, where data is dynamic and constantly evolving.

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
