# OpenReview forum: "Transfer Learning for Evolving Domains"
_TMLR — Under review for TMLR_

### Review · Reviewer_bjR7 · 2026-04-12

**Summary Of Contributions:**

# Summary of the paper

This paper proposes a new unifying framework for different "sub-problems" within domain adaptation, namely Transfer Learning for Evolving Domains (TrED). The authors positions different DA modalities within the "TrED continuum", going from unavailability of target domain data, to full availability of labeled data in that domain.

# Strong points

I think the main merit of this paper is establishing the continuum, which, as far as I know, is novel. This continuum is principled, in the sense that it reflects usual machine learning practice. I also think the section on foundation models to be insightful, as the interest in these kinds of models by the DA community is increasing.

# Weak points

I divide my criticism into 3 parts: **(i) The survey is not really pedagogical**; **(ii) Gaps in the TrED continuum**; **(iii) Failure to engage with broader literature**

## On the pedagogical value of this survey

While I think the TrED is an overall useful framework, the literature review (Sec. 3 and its subsections) feel a lot more like a list of existing work than an explanation of their core mechanisms, and how they fit into TrED. For instance, the only illustration of the discussed methods is Figure 2, and overall the paper has only 2 figures. In my opinion, **the authors should make an effort to illustrate the different methods they discuss within TrED**.

## Gaps in the TrED Continuum

There are 2 sub-problems in DA that are partially missing from the paper, namely Source-Free Domain Adaptation (SFDA) and Test-Time Domain Adaptation (TTDA). For instance, I would like to see the authors perspective on whether SFDA could be seen as a form of pre-training (pre-training on source, in that case), happening before domain generalization, in which case the adaptation part could happen at different stages after generalization, depending on whether labeled target domain data is available or not.

For TTDA, I particularly see it after multi-domain learning as a kind of "in-context learning mechanism". Here, labels can be available as well.

## Failure to engage with broader literature

There is a gap in the overall discussion of the paper, in the sense that the DA community already has methodologies for "evolving domains", namely, gradual domain adaptation [1, 2]. Another closely related concept is active/online learning [3]. Furthermore, though more on a speculative side, could In-Context Learning (ICL) be integrated into the TrED continuum as well?

[1] He, Yifei, et al. "Gradual domain adaptation: Theory and algorithms." Journal of Machine Learning Research 25.361 (2024): 1-40.

[2] Kumar, Ananya, Tengyu Ma, and Percy Liang. "Understanding self-training for gradual domain adaptation." International conference on machine learning. PMLR, 2020.

[3] Cacciarelli, Davide, and Murat Kulahci. "Active learning for data streams: a survey." Machine Learning 113.1 (2024): 185-239.

**Audience:**

Yes

**Audience Explanation:**

The paper is of great interest to the Domain Adaptation (DA) community, which is within the TMLR audience.

**Claims And Evidence:**

Yes

**Claims Explanation:**

The claims made in this paper are not empirical, which means that the "accurate, convincing and clear evidence" boils down to how well the paper articulates its main claim that,

> This survey presents a novel perspective on TL, viewing these previously isolated problems as a continuum. We discuss the evolution of data availability and examine TL methods designed for various stages of this process. As no single method currently addresses the entire spectrum, we highlight promising techniques that could potentially span a broader range of scenarios. By formalising this new perspective as a new task, Transfer Learning for Evolving Domains (TrED), this survey aims to establish a framework for research that will support the development of new TL methods that deal with the data availability continuum.

Which I think the authors do, especially by fitting existing work within the TrED continuum.

**Requested Changes:**

1, more critical) Include illustrations/figures and broader insights into the mechanisms of the different discussed methods and how they fit within the TrED continuum

2) Integrate SFDA and TTDA into the continuum (up to discussion, if the authors agree with my understanding)

3) Add additional discussion with respect gradual DA, continual learning and online domain adaptation, and how TrED differentiates itself from that case.

---

> ### Author Response · Authors · 2026-04-24
>
> We thank the reviewer for the quick and constructive feedback. We are working towards addressing the concerns raised and will provide a more detailed response once more reviews are submitted.

---

> ### Author Response · Authors · 2026-07-13
>
> We thank the reviewer for a constructive and encouraging review. Your request for discussing the relation between TrED and other learning paradigms, as well as a more critical and pedagogical discussion of the TL methods have guided some of the most structural changes in the new draft. We respond to these and your other comments below.
>
> 1. “On the pedagogical value”
> We agree that the submitted literature review reads as a list rather than an explanation, and two figures for a paper of this scope is too few. We think the underlying cause is structural: the literature section was organized by subproblem (DG, then UDA, then SDA, then MDL), which meant the same technique family reappeared in up to four separate places. Now, the section is being reorganized around types of technique, such that each family can be discussed once, in terms of its underlying mechanism, its behaviour across data-availability regimes, and its specific limitations for TrED. Also, by reframing the paper as a position paper rather than a survey, we buy space to discuss the different types of techniques with more depth. This includes adding figures illustrating the mechanisms of the technique families within the TrED continuum, showing how their behaviour changes across the data-availability regimes.
>
> 2. Gaps in the continuum: SFDA and TTDA
> We thank the reviewer for raising attention to these two domain adaptation sub-problems, because they are indeed closely related to TrED. However, as we are now highlighting in the revised paper, these problems are imposing constraints that are orthogonal to the TrED continuum. Namely:
> - SFDA is characterized by a constraint on source access at adaptation time: the source data is unavailable, there is only a source-trained model. But the target-side data availability in SFDA is that of UDA: only unlabelled target data. Meanwhile, TrED describes how the target data availability is changing over time. So SFDA is best understood not as a new stage of TrED, but as UDA under an additional access constraint.
> - TTA/TTDA is characterized by a constraint on when and how updates may occur: adaptation happens at inference time, on the test stream, typically without labels and under a single-pass or online restriction. Again, this is a restriction on the update protocol, not a new data-availability regime. Regarding your intuition that TTDA sits "after MDL as a kind of in-context learning mechanism", it may be pointing at the fact that the mature-domain regime is where a system has the richest prior knowledge to condition on. But we would locate the ICL connection with foundation models (where we do discuss it) rather than with TTA specifically, since TTA's defining feature is the online, label-free update constraint rather than the richness of the surrounding knowledge.
> We are expanding the discussion comparing these and other related problems with TrED in the new Section 3.5, and we hope that the added formalization of the learning protocol and evaluation metrics make these distinctions more clear.
>
> 3. Failure to engage with broader literature (gradual DA, active learning, online learning, and ICL)
> Similar to the previous point, the problems and techniques mentioned are indeed related to TrED, and we thank the reviewer for surfacing the relation, especially in regards to gradual DA which was an important omission in the submitted draft. These are also being discussed in the new Section 3.5, but in short:
> - Gradual Domain Adaptation (GDA) shares the TrED intuition that domains evolve over time, but in GDA the domain itself is shifting: the data distribution drifts through a sequence of intermediate domains, and the methodological insight is that self-training along that path can bridge a shift too large to cross directly. In TrED, the target domain may be fixed; what evolves is the availability of data and labels from it. These are genuinely different axes, and a system could face either without the other.
> - Active Learning treats label acquisition as a control variable that the learner can optimize (actively choosing which instances to request a label), while TrED treats label availability as an exogenous process that the learner must cope with.
> - Online Learning and Continual Learning share the streaming and sequential-update aspects, but not TrED's joint focus on multi-source transfer under evolving label availability.
> - ICL, as you suggest speculatively, does integrate naturally, and we develop this in the expanded FM section: ICL is a strong candidate mechanism for the SDA-like regime, since it exploits a small number of labelled target examples without a parameter update.

---

### Review · Reviewer_EZAh · 2026-05-18

**Summary Of Contributions:**

This paper proposes a new perspective for viewing traditionally isolated transfer learning problems — including Domain Generalisation (DG), Unsupervised Domain Adaptation (UDA), Supervised Domain Adaptation (SDA), and Multi-Domain Learning (MDL) — under a unified continuum based on evolving target-domain data availability. The authors refer to this perspective as Transfer Learning for Evolving Domains (TrED). The paper also provides a broad survey of transfer learning methods from the perspective of how they fit into different stages of this continuum.

**Strengths**

- The proposed TrED perspective is conceptually interesting and practically motivated.
- The survey is comprehensive, covering a large number of methods across several transfer learning subfields.

**Weakness**

- The proposed TrED framework remains largely conceptual, with limited concrete technical synthesis or evaluation criteria for improving the system as a whole.
    - From the methodological perspective, while the paper discusses how certain techniques may connect to TrED, the discussion mainly remains at the level of describing existing components within each subfield. The paper does not clearly identify what additional designs are needed to build a unified TrED solution.
    - The paper does not clearly define how a TrED system should operate and be evaluated over time. In particular, it is unclear how transitions between stages should be detected or handled in practice, e.g., when a system should behave more like DG, UDA, or SDA. Similarly, the paper lacks discussion of holistic evaluation protocols for TrED systems.
- The survey sometimes becomes overly enumerative, listing many methods without sufficiently deep comparative analysis.

**Audience:**

Yes

**Audience Explanation:**

This paper provides a fresh perspective on transfer learning and may serve as a useful survey and conceptual problem formulation for researchers working on domain generalization, domain adaptation, continual learning, and related areas.

**Claims And Evidence:**

No

**Claims Explanation:**

The formulation is conceptually interesting and supported by several motivating examples. However, the paper does not sufficiently establish what new algorithmic challenges or technical requirements uniquely arise from the TrED setting beyond reorganizing existing transfer learning problems under a shared narrative.

**Requested Changes:**

As pointed out in the weakness, the paper would benefit from

- clarifying the concrete added value of TrED beyond organizing existing DG/UDA/SDA/MDL settings along a temporal data-availability axis.
- a more systematic discussion of what properties a TrED system should satisfy and how it should be evaluated experimentally (e.g., robustness during stage transitions, long-term performance)

---

> ### Author Response · Authors · 2026-05-28
>
> We thank the reviewer for the constructive feedback. We have started to address some of the comments, and will provide a more detailed response once more reviews are submitted.

---

> ### Author Response · Authors · 2026-07-13
>
> We thank the reviewer for a concise and well-targeted review. Your central point (i.e. that TrED remains largely conceptual and does not yet establish what new technical requirements arise beyond reorganizing existing settings) is one we take seriously, and it is closely aligned with concerns raised by the other reviewers. It has driven the main revision described in our global comment: reframing the paper as a position paper that formalizes TrED as an open problem. We respond to each of your weaknesses below.
>
> 1. "The framework remains largely conceptual, with limited concrete technical synthesis or evaluation criteria."
> We agree that the submitted version left some aspects of TrED under-specified, namely the learning protocol and evaluation metrics. We provide a formal description for both in this revision.
> - Regarding the learning protocol, given the data-availability evolution of a target domain, a solution chooses a sequence of update times at which it may access the currently available labelled and unlabeled datasets and produce an updated model by minimizing a per-step loss. The update schedule and the per-step loss are design decisions of the method. The problem fixes only the data-availability process (including the label-delay mechanism) and the evaluation criterion.
> - Regarding evaluation criterion, a TrED solution is scored by the cumulative target risk of its currently active model, rather than by a single risk measure evaluated on the final model version. We also discuss complementary metrics (density weighting to avoid over-counting high traffic windows, and performance at fixed data requirements) and the added difficulty of estimating current time performance in environments with label delay.
>
> 2. "The paper does not identify what additional designs are needed to build a unified TrED solution."
> The revised Section 4 is organized to answer exactly this question. It begins with the stage-switching baseline in Section 4.1 (select DG when no target data exist, UDA once unlabeled data arrive, and so on), which is the most obvious candidate for a unified solution, and identifies concretely where it falls short: the seam costs incurred at regime transitions; the difficulty of choosing when to switch in a label delay setting; and the assumption that the optimum decomposes into discrete per-stage methods, which need not hold. It then examines existing technique families for mechanisms that could support a more holistic solution in Sections 4.2 and 4.3, and closes with foundation models as the strongest existing candidate in Section 4.4, while identifying what they still do not address (transition dynamics, exploitation of label-delay structure, optimization of a cumulative rather than snapshot objective, and evaluation leakage).
>
> 3. "It is unclear how transitions between stages should be detected or handled — when a system should behave more like DG, UDA, or SDA. The paper lacks holistic evaluation protocols."
> On the evaluation half, the cumulative-risk criterion described above is precisely the holistic protocol you ask for: it scores the active model across the whole trajectory, which is what makes properties like robustness during transitions and long-term performance (both of which you name in your requested changes) into measurable quantities rather than qualitative aspirations. A method that degrades at a regime boundary is charged for it; a method that plateaus late is charged for that too.
> On transition detection, we think the question contains an assumption worth noting. Asking when a system should switch to behaving like DG, UDA, or SDA presupposes that a TrED solution is composed of discrete stage-specific behaviors between which it must arbitrate. That is a property of the switching baseline and is an open problem for that type of TrED solution (that we identify and leave for future work), not a property of the TrED problem itself. A solution whose behavior varies continuously with the available data need not detect transitions, because there are no discrete modes to switch between. We now make this explicit: the stage boundaries are naturally diffuse, which is one of the arguments for TrED as a distinct problem. We are expanding this discussion in the new Section 4.1.
>
> 4. "The survey sometimes becomes overly enumerative, listing many methods without sufficiently deep comparative analysis."
> Agreed, and this is one of the changes we are most confident improves the paper. The literature review is being reorganized into Section 4.2 around types of technique rather than around the four sub-problems. Under the old structure, the same technique family reappeared in four separate places (once per sub-problem), which mechanically produced enumeration and suppressed comparison. Organizing by technique brings those instances together, so each family can be discussed once, in terms of its underlying mechanism, its behavior across data-availability regimes, and its specific limitations for TrED.

---

> > ### Author Response · Authors · 2026-07-13
> >
> > On your requested change.
> > You asked us to clarify TrED's concrete added value beyond organizing DG/UDA/SDA/MDL along a temporal axis, and to systematically discuss what properties a TrED system should satisfy and how it should be evaluated. Our contribution is the problem specification of Section 3: a formal description of the natural evolution in data availability, the learning protocol that exposes where a method's design decisions live, and a cumulative target risk and other evaluation metrics to assess the predictive performance. The properties you name (robustness during stage transitions and long-term performance) follow from the evaluation procedure, and the goal of the TrED solution can be described as: what policy minimizes cumulative risk across the whole trajectory? That question, we argue, is new, well-posed, and unsolved.

---

### Review · Reviewer_jDTx · 2026-06-29

**Summary Of Contributions:**

## Review summary

This paper proposes Transfer Learning for Evolving Domains (TrED), a perspective in which standard transfer-learning settings, namely Domain Generalization (DG), Unsupervised Domain Adaptation (UDA), Supervised Domain Adaptation (SDA), and Multi-Domain Learning (MDL) are interpreted as stages along a data-availability continuum. The motivating scenario is that a new target domain initially has no data, then accumulates unlabelled data, then labelled data, and eventually becomes a mature domain with enough labels to be treated as another source/domain in a multi-domain learning setup. The paper formalizes this idea using time-indexed datasets with observation timestamps and label-availability timestamps, then surveys transfer-learning methods according to where they fall in this proposed continuum. This can be considered a strength of the paper, as it is trying to propose a novel way of looking at multiple problems. Nevertheless, I find the central contribution insufficiently convincing in its current form. The proposed TrED perspective currently feels more like a relabelling of existing static transfer-learning settings than a genuinely new problem formulation. The paper’s own literature review reinforces this concern: only a small number of methods actually span multiple stages, while most of the paper becomes a conventional survey of transfer learning methods grouped under a new terminology.

## Strengths

- The paper starts with an important practical consideration, which is that transfer-learning assumptions are often static, whereas real systems evolve as data and labels are collected over time. The time-indexed notation for data and label availability offers a new perspective on the distinction between when features are observed and when labels become available. Figure 1 is an effective visualization of the aforementioned cycle.

- The authors have correctly recognized that foundation models are highly relevant to this discussion. Their ability to support zero-shot prediction, pseudo-labelling, and in-context learning ability makes them arguably the most natural candidates for transfer across multiple data-availability regimes.

## Weaknesses

- The main weakness is that the proposed perspective does not yet establish a genuinely new technical problem. The paper observes that DG, UDA, SDA, and MDL can be placed on a continuum indexed by target-domain data and label availability. However, this observation alone is not enough to define a new research problem. A straightforward baseline would be to select a method based on the current timestamp or current data-availability regime: use DG when no target data exist, use UDA once unlabelled target data exist, use SDA once labels arrive, and use MDL once enough labelled target data are available. The paper does not explain why this switching strategy is inadequate, nor does it formulate what a unified TrED method should optimize beyond avoiding manual method changes. The paper claims that a solution should “operate continuously” across the spectrum but this objective is not made precise enough to distinguish TrED from a pipeline that simply switches methods at transition times.

- The formalization is biased toward classification. For example, the discussion in Section 3.1.1 implicitly treats many transfer-learning difficulties as classification-specific. In particular, the discussion of target label spaces and open-set recognition makes sense for classification, but it does not transfer cleanly to regression. For regression, the analogous issues would be conditional shift or extrapolation outside the observed target support. These are not the same as open-set recognition. The authors should either restrict the scope explicitly to classification-heavy transfer learning or revise the formalization and survey taxonomy to treat regression on equal footing.

- The category “Data Transformation” groups together quite different mechanisms: augmentation, feature mapping, instance weighting, kernel alignment, and pseudo-labelling. This makes the taxonomy less informative than it could be. For example, DDAIG is better understood as a data augmentation method rather than a generic data-transformation method. Meanwhile, DICA is naturally connected to the broader family of kernel alignment and invariant representation learning methods. Placing both under a broad “data transformation” heading obscures the technical distinctions that matter.

- The paper presents itself as a survey of the state of the art, but the coverage is uneven. Some sections list only a small number of methods despite existing literature around domain adaptation, kernel alignment, invariant learning, test-time adaptation, and representation learning for distribution shift. This is problematic because the paper makes broad claims about what current work does or does not address. Without a clear survey methodology, it is hard to know whether the observed gaps are real or simply artifacts of the paper-selection process. The authors should include a review protocol: databases searched, keywords, date range, inclusion/exclusion criteria, number of papers considered, number retained, and how methods were categorized. Finally, Tables 2 and 3 use numbered references while the text uses surname and year, which renders the Tables useless.

- Some of the discussion lacks technical precision. For example, Section 3.1.3 states that perturbation-based methods are “probably not general enough.” This is not a rigorous claim. What does “general enough” mean? The paper often gestures toward limitations but does not give proper or intuitively compelling motivations for them.

- Section 3.5 undermines the paper’s main thesis. It states that only a small part of the reviewed literature addresses more than one stage of the proposed TrED continuum. This observation is important, but it creates a structural problem for the paper. If the literature barely contains methods that span multiple stages, then the paper is not yet a survey of TrED methods. It is mostly a survey of transfer-learning methods, with a relatively thin TrED interpretation layered on top. The current framing of the paper thus oscillates between “we introduce a new perspective” and “we survey the state of the art,” but the latter is not well supported.

- The foundation-model section may be the most relevant part of the paper, but it is severely underdeveloped. Foundation models are arguably the clearest existing attempt to cover the complete transfer-learning spectrum through scale: zero-shot generalization for the DG-like stage, prompting or pseudo-labelling for UDA-like settings, in-context learning or parameter-efficient fine-tuning for SDA-like settings, and retrieval/fine-tuning/adapters for mature multi-domain use. This raises a major question that the paper does not adequately address: are foundation models already the most plausible instantiation of TrED?

**Audience:**

Yes

**Audience Explanation:**

Transfer learning is a big part of the machine learning literature and a very important problem in practical applications. The paper's topic falls within the scope of TMLR.

**Claims And Evidence:**

No

**Claims Explanation:**

See weaknesses section in the first part of the review.

**Requested Changes:**

The current submission is not fit for acceptance to TMLR even with a major revision, as most of my listed weaknesses would effectively transform it into a new article. Nevertheless, I believe that the following points would improve the manuscript for the future.


1. Clarify whether the contribution is a survey, a position paper, or a problem formulation. In its current form, it is closest to a position paper plus a broad transfer-learning survey.

2. Formalize TrED as an actual learning protocol. Define the model update times, label-delay process, and possible training objectives function. The paper should make it clear why TrED is not solved by timestamp-based method switching.

3. Add a rigorous survey methodology, including sources and why papers were included, how they were screened etc, if the survey style is what the authors believe to be the right. Without this, the literature coverage and “no current work” claims remain difficult to trust.

4. Revise the taxonomy. Separate data augmentation, feature mapping, kernel alignment, instance weighting, pseudo-labelling, normalization adaptation, and generative translation. The current “data transformation” category is too broad.

5. Either restrict the scope to classification only problems or add treatments on regression. The discussion as of now relies mainly onclassification-specific concepts.

6. Expand the foundation-model section substantially. This section should become one of the main contributions of the paper, because FMs are the strongest existing example of a system that may operate across the full transfer-learning spectrum proposed here.

7. Try to propose a benchmark protocol.

The following question might help the authors in directly applying the above points to specific parts of the paper stronger

a. Why is TrED not adequately solved by switching among DG, UDA, SDA, and MDL methods as target data and labels become available?What is the objective of a TrED algorithm: final target risk, cumulative target risk, regret against an oracle switching policy, compute-aware risk, or multi-domain risk?

b. What was the methodology for selecting the surveyed papers?

c. How does DICA relate to the broader kernel alignment and invariant representation learning literature?

d. What literature is intentionally excluded, especially from source-free adaptation, test-time adaptation, active learning, online transfer learning, and continual domain adaptation?

e. Are foundation models the most natural realization of TrED? If not, what essential aspect of TrED do they fail to capture?

f. What baseline would the authors recommend for a simple stage-wise method-switching policy?

g. What benchmark would allow the community to measure progress on TrED?

---

> ### Author Response · Authors · 2026-07-13
>
> We thank the reviewer for an exceptionally detailed and constructive review. Incorporating your feedback will significantly improve the quality of the paper. We respond to each requested change below, then to your questions (a–g), which we found very useful for targeting the revision.
>
> 1. Clarify the contribution (survey / position paper / problem formulation).
> We are reframing the paper as a position paper whose primary contribution is the formulation of TrED as an open problem. The revised paper no longer makes a systematic-survey claim; the literature discussion becomes a targeted diagnostic of which existing techniques are promising for TrED and where they fall short (see change 3). This directly resolves the "oscillation" you identify between "new perspective" and "survey."
>
> 2. Formalize TrED as a learning protocol; make clear why timestamp switching does not solve it.
> We now provide a formal description of the learning protocol of a TrED solution. Given the data-availability evolution of a target domain, a solution chooses a sequence of update times at which it may access the current labelled and unlabeled datasets and produce an updated model by minimizing a per-step loss. The update schedule and the per-step loss are design decisions of the method. The problem fixes only the data-availability process (including the label-delay mechanism) and the evaluation criterion.
> For evaluation, we make the objective precise: a TrED solution is scored by the cumulative target risk of its currently active model, rather than by a single risk measure evaluated on the final model version. We also discuss complementary metrics (density weighting to avoid over-counting high traffic windows, and performance at fixed data requirements) and the added difficulty of estimating current time performance in environments with label delay.
> After formalizing the learning protocol and evaluation metrics, we can state precisely why timestamp-based switching is indeed one possible type of solution, but one with shortcomings that a TrED solution should aim to address:
> When transitioning from one data availability regime to the next, there can be a period of time where neither solution is optimized to the specific setting. For example, if there is a small unlabeled target domain dataset available (between DG and UDA regime), a DG solution will typically not leverage it, while it may be insufficiently large to be used by a UDA solution. And the cumulative target risk metric makes this seam cost measurable. As such, it is possible that other TrED solutions with a more holistic view of the continuum can be further optimized to improve performance in these regions.
> Choosing when to switch is itself hard under label delay: the signal that would tell you a transition has occurred (i.e. the target domain performance) may be inaccessible at decision time. The switcher must commit to switch timing without the labels that would justify it.
> The switching formulation presupposes that the optimal solution decomposes into discrete per-stage methods. It need not: the optimum may be a continuous policy that no finite switching pipeline expresses.
> As such, the stage switcher is a reasonable solution for the TrED problem. But we argue that this is not THE solution, but rather a greedy, oracle-dependent solution, with some potential limitations that we highlighted before. Beating it is a well-posed and, to our knowledge, unsolved problem, which is precisely the open problem TrED names. We think stating this honestly is stronger than over-claiming a resolution, and we dedicate the new Section 4.1 to discuss this in detail.
>
> 3. Add a survey methodology, or drop the survey claim.
> We drop the survey claim and we do not assert exhaustive coverage. The new Section 4.2 states plainly that it examines representative techniques to identify TrED-promising mechanisms and their limitations, not to survey the field. We considered adding a full review protocol instead, but concluded it would pull the paper back toward the survey framing that generated most of your concerns; the position-paper route is the more coherent choice.

---

> > ### Author Response · Authors · 2026-07-13
> >
> > 4. Revise the taxonomy; "Data Transformation" is too broad.
> > While restructuring the literature review in the new Section 4.2, we are revisiting the taxonomy as well, and clarifying our reasoning. Our goal with the “Data Transformation” category is to aggregate different techniques that affect the data in order to improve transfer performance. Together with “Model Adaptation” (changes to the model architecture) and “Training objective” (changes to the metric that is being optimized), these should be read as broad categories that map TL techniques along different (somewhat independent) axes. However, we agree that this may not have been clear in the submitted draft. As such, we are clarifying the reasoning behind the taxonomy and we retain the three top-level categories as axes, but promote the technique-level distinctions from table rows to the organizing units of the text. By describing the literature in terms of groups of techniques rather than types of problems, we aim to draw more attention to their similarities and trade-offs.
> > Regarding the specific examples that you mentioned: we already categorize DDAIG as a data augmentation; and DICA as a feature mapping, alongside other distribution-alignment methods such as CORAL (Sun et al., 2016) and subspace alignment (Saenko et al., 2010). We then include both within the broader category of Data Transformation. We agree that the presentation in the submitted may have buried this distinction, and we strive to make this type of categorization clearer by following the taxonomy of techniques rather than the problems, and by fixing the formatting of the tables.
> >
> > 5. Restrict scope to classification, or treat regression on equal footing.
> > The core TrED formalism (which includes the data-availability process, the learning protocol, and the cumulative target risk) is defined over generic predictive tasks and losses, and is agnostic to classification vs. regression. The classification bias you correctly flag is localized: the passage in old Section 3.1.1 that presents open-set recognition as a general transfer difficulty. You are right that this does not transfer to regression, where the analogous concerns are conditional shift and extrapolation beyond the observed target support. We are correcting that passage and adding a sentence stating the formalism's neutrality. We will be transparent that the illustrative examples still lean on classification tasks, as that is where the cited literature concentrates, but the formalism itself does not.
> >
> > 6. Expand the foundation-model section into a main contribution.
> > We agree FMs are the strongest existing candidate for the basis of a complete TrED solution and we are expanding this in the new Section 4.4. We are mapping its capabilities onto the data availability regimes (zero-shot/prompting for DG; prompting/pseudo-labeling/distillation for UDA; ICL and fine-tuning for SDA; and RAG or adapters for MDL), while also highlighting what they do not yet address as TrED solutions: transition dynamics between data regimes, how to better exploit the label delay structure, how to optimize the cumulative objective, and the evaluation leakage risk problem.
> >
> > 7. Propose a benchmark protocol.
> > The evaluation criterion in change 2 describes the metric that a benchmark needs: it defines what to measure (cumulative target risk under an evolving, label-delayed data-availability process). We specify this primitive and the properties a benchmark must have (an explicit data-availability schedule, a label-delay simulation, and scoring of the active model over time). We are not presenting a concrete dataset suite with fixed splits, and flag it as future work.

---

> ### Author Response · Authors · 2026-07-13
>
> (a) Why not switch between solutions? What is the objective?
> The objective is cumulative target risk. Regret against an oracle switching policy seems like a natural alternative, but doing so would build the discrete-decomposition assumption into the problem definition, which is the very assumption we argue a TrED solution may need to violate. As such, the switching strategy is one example of a possible solution, with potential limitations. See change 2.
> (b) Survey methodology?
> We drop the survey claim rather than add a protocol. See change 3.
> (c) DICA categorization
> DICA is described as a feature mapping technique, within the data transformation type. See change 4.
> (d) Excluded literature
> The referred problems are being directly discussed in Section 3.5 of the new draft. Briefly: SFDA and TTA differ from TrED on an orthogonal axis (source-access constraints and offline-vs-online update timing) rather than being further points on the data-availability continuum; continual DA and online TL share the streaming aspect but not TrED's joint focus on multi-source transfer under evolving label availability; active learning is related through the label-acquisition angle but treats acquisition as a control variable rather than an exogenous process.
> (e) Are FMs the natural realization?
> They are the most complete basis, but not yet a solution. See change 6.
> (f) Recommended stage-switching baseline?
> We describe the DG->UDA->SDA->MDL stage-switching strategy as a viable type of solution to TrED and discuss its limitations in the new Section 4.1. We do not propose any switching time rule, and highlight it as an open problem.
> (g) Benchmark?
> We propose the cumulative risk and other performance metrics as measurement primitives, and leave the dataset suite and splits for future work. See change 7.

---

### Author Response · Authors · 2026-07-13
**Official Comment by Authors — Summary of Planned Revisions**

We thank all three reviewers for their careful and constructive reading. Before responding to each review individually, we want to describe the main revisions that we are implementing, as they address the main concerns that multiple reviewers raised.

We wrote this paper with two main goals: first, to present TrED as a novel perspective of TL inspired by real-world applications, taking into account the typical evolution of data availability over time; and second, to conduct a literature review focused on the most promising TL methods to solve TrED, their limitations, and open problems.

However, several comments from multiple reviewers stem from the duality of these goals. On one hand, reviewers EZAh and jDTx requested a more extensive description and discussion about the TrED problem, including a formal description of the training procedure and evaluation metrics, with a clearer discussion on how this problem differs from other TL and ML paradigms. On the other hand, all reviewers found the literature review insufficiently critical (with reviewer EZAh describing it as “overly enumerative”) or potentially incomplete (with reviewer jDTx requesting that we include the review protocol to support some of our claims).

Following reviewer jDTx‘s recommendation, we decided to reframe the work as a position paper that primarily proposes TrED as an open problem, with a more focused but more critical analysis of relevant prior work. Concretely, the revision makes three structural moves:
- First, TrED is formalized as a standalone problem, motivated by the typical data availability evolution. Then, the four TL sub-problems (DG, UDA, SDA, MDL) are shown to be specific instantiations of the TrED problem at different points along this continuum. We also dedicate a subsection to distinguishing TrED from other related ML problems (including SFDA, TTA, and GDA, as proposed by reviewers bjR7 and jDTx). We hope this clarifies why we believe TrED is a new and relevant problem to be addressed.
- Second, we provide a formal description of the learning protocol for a TrED solution, and discuss several options for evaluation metrics (as requested by reviewers EZAh and jDTx). This formalizes the training and evaluation methods, while opening the path for future benchmark studies.
- Third, we clarify the scope of our literature review, framing this section as an exploration of paths towards a TrED solution. We begin with the stage-switching baseline proposed by reviewer jDTx, and argue why TrED is not reducible to timestamp-based method-switching. Then, we examine which existing TL techniques are most promising for a unified solution and where they fall short. Importantly, we reorganize this section around types of techniques, rather than presenting it as a systematic review of the four TL sub-problems. This allows us to have a more critical and targeted discussion around the relevant methods for TrED, while avoiding the risk of making unsupported claims (as flagged by reviewer jDTx).

Given the many points of reframing the work, we did not manage yet to complete the new draft and we are presenting the planned changes in this message, with further more detailed discussions on the per-reviewer responses. We will share the revised manuscript before the end of the week (Friday 17th of July). Meanwhile, we are grateful for the feedback, which will significantly improve this contribution.

---

> ### Author Response · Authors · 2026-07-13
>
> While the revised manuscript is not yet ready to share, we are posting the new table of contents here so that reviewers have a high-level view of the paper's structure while reading our individual replies. We hope this makes the scope and effect of the reframe concrete, and shows where each of the requested changes lands.
>
> 1. Introduction
> 2. Background
> 2.1 Definitions
> 2.2 Classical TL settings
> 3. TrED
> 3.1 Evolution of data availability
> 3.2 Learning protocol
> 3.3 Evaluation
> 3.4 Classical TL settings as regimes of TrED
> 3.5 Relation to other learning paradigms
> 4. Paths to a TrED solution
> 4.1 The stage-switching baseline
> 4.2 Building blocks from the TL literature
> 4.3 Methods spanning multiple regimes
> 4.4 Foundation models
> 5. Conclusions
>
> Three structural points are worth highlighting. First, TrED is now developed as a standalone problem in Section 3, with the learning protocol (3.2) and evaluation criterion (3.3) given dedicated treatment, and the four classical settings recovered as regimes of the formalism (3.4) rather than serving as the paper's organizing skeleton. Second, Section 3.5 is new, and situates TrED against neighbouring paradigms (online and continual learning, online transfer learning, TTA, SFDA, and gradual domain adaptation). Third, the literature discussion (Section 4) is reframed as an exploration of paths toward a TrED solution: it opens with the stage-switching baseline (4.1) and is then organized by type of technique (4.2) rather than by subproblem, which is what allows a more critical and comparative treatment.